# Detection of *Rhodococcus fascians*, the Causative Agent of Lily Fasciation in South Korea

**DOI:** 10.3390/pathogens10020241

**Published:** 2021-02-20

**Authors:** Joon Moh Park, Jachoon Koo, Se Won Kang, Sung Hee Jo, Jeong Mee Park

**Affiliations:** 1Forest Resource Research Division, Jeollabuk-do Forest Environment Research Institute, Jinan 55454, Korea; joonmoh@korea.kr; 2Division of Science Education and Institute of Fusion Science, Jeonbuk National University, Jeonju 54896, Korea; jkoo@jbnu.ac.kr; 3Biological Resource Center, Korea Research Institute of Bioscience and Biotechnology, Jeonguep 56212, Korea; bioksw@kribb.re.kr; 4Plant Systems Engineering Research Center, Korea Research Institute of Bioscience and Biotechnology, Daejeon 34141, Korea; shrrd@kribb.re.kr

**Keywords:** fasciation, *fasD*, lily, *Rhodococcus fascians*, *vicA*

## Abstract

*Rhodococcus fascians* is an important pathogen that infects various herbaceous perennials and reduces their economic value. In this study, we examined *R. fascians* isolates carrying a virulence gene from symptomatic lily plants grown in South Korea. Phylogenetic analysis using the nucleotide sequences of 16S rRNA, *vicA*, and *fasD* led to the classification of the isolates into four different strains of *R. fascians*. Inoculation of *Nicotiana benthamiana* with these isolates slowed root growth and resulted in symptoms of leafy gall. These findings elucidate the diversification of domestic pathogenic *R. fascians* and may lead to an accurate causal diagnosis to help reduce economic losses in the bulb market.

## 1. Introduction

The genus *Rhodococcus* contains Gram-positive aerobic bacterial species belonging to the phylum Actinobacteria. These bacteria can be isolated from water, soil, and marine habitats, including harsh ecological regions such as the Arctic, deserts, and heavily polluted areas [1,2]. Interest in *Rhodococcus* species has been increasing because of their biotechnological applications and potential in bioremediation [3,4]. Most *Rhodococcus* species grow epiphytically or endophytically in the roots or leaves of plants, are nonpathogenic, and are beneficial, aiding the host plant in environmental adaptation [5,6,7].

To date, only two members of the genus are known to be pathogenic: the plant-pathogenic *Rhodococcus fascians* and livestock-pathogenic *Rhodococcus equi* [8,9,10]. Pathogenic *R. fascians* causes fasciation and leafy gall disease in a wide spectrum of plants, including 87 genera spanning 40 plant families [10,11,12]. The growth abnormalities and morphological alterations caused by this infection result in severe economic losses in ornamental horticulture industries in many countries. In addition to leafy gall, infected plants exhibit symptoms of witches’ broom and other deformed growths [12], similar to those of plants with *Agrobacterium* infection or under abiotic stress [13,14].

Pathogenic *R. fascians* is often identified by its phenotypic characteristics in selection medium [15,16]; by molecular identification of virulence plasmids via PCR [17,18]; or by observation of disease symptoms in host plants, such as *Nicotiana benthamiana* [19]. Pathogenic *R. fascians* contains the linear plasmid pFiD188 with *fas*, *att*, and *hyp* virulence loci, and the cytokine production causes disease symptoms [20,21,22]. The only known virulence gene present on the chromosome of *R. fascians* is the *vicA* gene, which encodes malate synthase, an enzyme in the glyoxylate cycle [23]. However, *R. fascians* isolates without virulence genes are often cultured from symptomatic tissues [18,24].

In lily bulbs imported to South Korea, *R. fascians* has been detected as a result of strict domestic post-entry quarantine measures. A large proportion of these bulbs, the second most traded in South Korea, were thus destroyed, which became a social issue in South Korea [25]. Although not very large, the South Korean flower bulb market has been expanding steadily since 2016 and is regarded as a stable growing market among Dutch flower bulb exporters [26].

In this study, we aimed to identify the causative agent of fasciation in lily plants grown in Gangwon province, South Korea. The four identified *R. fascians* strains carried known virulence genes, such as *fasD* and *vicA*, and their infection caused disease symptoms in *N. benthamiana*. Overall, the accurate identification and characterization of *R. fascians* strains that induce lily fasciation will be helpful in the control and epidemiological investigation of this pathogen in South Korea.

## 2. Results and Discussion

### 2.1. Detection of R. fascians Virulence Gene in Symptomatic Lily Plants

To determine the causal agent of fasciation in lilies, eight hybrid symptomatic plants (YWS1 to YWS8), including Asian and Oriental lily varieties, were sampled from greenhouses and nurseries (Figure 1, Appendix A). These plants exhibited fasciated flat stems, multiple-shoot proliferation, and stunted growth indicative of symptoms caused by *R. fascians* infection [12]. Figure 1B shows a plant with severe stunted growth and flat twisted stems. To verify infection by *R. fascians*, the total bacterial genomic DNA isolated from the plant samples was analyzed for the presence of the virulence gene *fasD* by PCR (Appendix A, Figure 1C). *fasD* is present in the *fas* operon in the pathogenic linear plasmid of *R. fascians* D188 [20] and encodes isopentenyl transferase, which is a major enzyme involved in cytokine synthesis that is widely used as a diagnostic marker for the pathogenicity of *Rhodococcus* species [17,18]. Sequencing of the amplified band revealed 100% identity with the *fasD* sequence of pathogenic *R. fascians* strain D188. These results suggest that the cause of the disease symptoms in these lilies was infection with pathogenic *R. fascians*.

### 2.2. Development of Genome-Based PCR Primers for Detection of Pathogenic R. fascians

Both plasmid- and chromosome-mediated virulence genes are involved in the pathogenicity of *R. fascians* [19]. We detected the same *fasD* gene in all symptomatic lily samples and therefore tried to develop chromosome-based pathogenic markers for the accurate screening of causative bacteria. We targeted *vicA*, the only virulence gene present on the chromosome. However, there are reports that conventional *vicA* primers (VicA1497F and VicA1990R) [18] have problems in diagnosing phytopathogenic *Rhodococcus* species [20]; therefore, we decided to develop new *vicA*-based diagnostic primers. To this end, we obtained the full sequence of *vicA* from 50 *Rhodococcus* species including five pathogenic *R. fascians* strains from publicly available sources and identified variable regions that distinguish pathogenic species (Appendix A), for which new primers were designed (vicA44-F and vicA737R) (Appendix A). The detection efficacy of these primers was tested via PCRs using genomic DNA of a subset of 10 *Rhodococcus* species as templates (Figure 2A). DNA from purchased *R. fascians* (Loewe Biochemica GmbH, Sauerlach, Germany) was used as a positive control. No amplification with these primers was observed for the DNA of the avirulent *Rhodococcus* species, but amplification was detected for the positive control. Additionally, amplification was detected in symptomatic lily specimens but not in asymptomatic lilies (Figure 2B). These results are consistent with *fasD*-based detection, demonstrating the reliability of the newly designed *vicA*-based primers for *R. fascians* detection. As more bacterial genomic sequences become available, additional regions may be revealed as suitable targets for the diagnosis of *Rhodococcus* species.

### 2.3. Isolation of R. fascians from Symptomatic Lily Plants

To isolate the bacteria from plant tissues, serial dilutions of root extracts from symptomatic lily plants were plated on semi-selective medium for *R. fascians*. As *R. fascians* has an orange-colored pigment [12], we initially selected the colored bacterial colonies. Orange colonies were found in four out of eight plant samples with varied disease symptom severities. Ten orange colonies were picked from each plant sample and cultured, and PCR was used to determine the presence of *vicA* and *fasD*. For nucleotide sequence analysis of the *fasD* gene, a new primer was synthesized and used instead of the conventional *fasD* diagnostic primer (Appendix A). It was capable of amplifying a larger fragment and detecting both known *fasD* genes (Appendix A). *vicA*-specific amplification was detected in 19 of 40 samples, and *fasD* was also detected in 12 of these samples (Appendix A). To investigate the genetic diversity of these 12 isolates and their relation to known pathogenic *Rhodococcus* species, *vicA*, *fasD*, and 16S rRNA genes were sequenced, and a phylogenetic analysis was performed (Appendix A). BLAST search results showed that all isolates had the highest homology with pathogenic *R. fascians* strains in terms of 16s rRNA genes (≥ 99.4% identity) and *fasD* (100% identity) (Table 1). In terms of *vicA* sequences, they showed 96.4–99.1% sequence similarity to D188. With nonpathogenic *Rhodococcus* KB6 as an outgroup [5], the phylogenetic analysis revealed that the 12 isolates were divided into four groups (Appendix A). We also performed a phylogenetic analysis of the *vicA* sequences from strains representing each of these four groups and 50 known *Rhodococcus* species (Figure 3). All four strains belonged to the same lineage as the pathogenic *Rhodococcus* species, and they were closer to D188 with the linear virulence plasmid than to A25f carrying virulence genes on its chromosome [19,22,24]. Two strains, YWS3-1 and YWS8-2, belonged to the same branch as D188 (Figure 3). These results suggest that YWS isolates from symptomatic lilies are phytopathogenic *R. fascians*.

Most plant pathogenic *Rhodococcus* have virulence plasmids; however, two exceptional strains, A21d2 and A25f, have been reported, which are presumed to have virulence genes in their chromosomes [24]. In particular, A21d2 lacks two-thirds of the *fas* operon, including *fasD*, which is replaced by a novel gene chimera in islands within the chromosome. Therefore, we investigated whether seven orange colonies that produced the pathogenic *R. fascians*-specific *vicA* amplicon, but not the *fasD* amplicon, are related to A21d2. A sequence comparison based on *vicA* showed that they all belong to the same clade as D188 (data not shown), suggesting that they lost their virulence plasmids during the screening process.

### 2.4. Pathogenicity of YWS Isolates on Tobacco Plants

To verify the pathogenicity of the isolated strains using *N. benthamiana* [19], we selected isolates YWS1-1, YWS3-1, YWS4-1, and YWS8-2 as representative strains according to the *vicA*-based phylogenetic analysis (Figure 3). Infection of *N. benthamiana* seedlings with YWS3-1 and YWS8-2 significantly stunted root growth (*p* ≤ 0.05), whereas inoculations with isolates YWS1-1 and YWS4-1 did not alter growth compared with that of mock-treated control seedlings (Figure 4A). Similarly, the meristems of tobacco plants inoculated with YWS3-1 and YWS8-2 exhibited the typical symptoms of leafy gall, including multiple proliferation of deformed leaves, foliar distortion, and generation of adventitious tissue (Figure 4B). On the other hand, YWS1-1 and YWS4-1 induced less severe symptoms than the other two strains, and they delayed growth and weakened the dominance of the shoot apex compared with mock-inoculated control plants. These symptoms were more pronounced for YWS1-1 than for YWS4-1 (Figure 4B). The bacteria were re-isolated from symptomatic *N. benthamiana* plant tissues, and *vicA* PCR analysis confirmed that they were the same bacteria as the inoculated bacteria (data not shown).

Differences in the severity of disease symptoms caused by *R. fascians* D188 are related to the level of expression of the *fas* operon present in the virulent plasmid [20,21,22]. Although we have not yet observed *fas* transcripts expressed by the four isolates, the symptomatic difference in *N. benthamiana* may be due to differences in *fas* expression in these isolates. Another possibility is that the differences in symptom severity might be related to differences in the virulence genes present in their genomes. Indeed, nonpathogenic *R. fascians* can alter root structure, including the proliferation of root hairs and early lateral roots, without retarding root growth [20]. However, inoculations with YWS1-1 and YWS4-1 did not delay root growth, and we did not observe any deformation of roots (Figure 4A). Therefore, we propose that the two isolates are D188 variants with the same virulence plasmid as D188, but with different pathogenicity due to differences in chromosomal virulence determinants rather than being nonpathogenic.

In conclusion, we demonstrated for the first time that the causative agent of the fasciated lily plant grown in Gangwon, South Korea, is phytopathogenic *R. fascians*. Publicly available whole genome sequence information of *Rhodococcus* species allowed us to develop diagnostic primers specific to pathogenic *R. fascians* by targeting the virulence gene *vicA* on the chromosome (Figure 2). In combination with the plasmid-based virulence marker *fasD*, we confirmed that the bacteria isolated from symptomatic lilies were pathogenic *R. fascians.* Furthermore, the isolated pathogenic *R. fascians* was classified into four strains based on the sequence diversity of the pathogenic *vicA* amplicon (Figure 3). This classification correlated with differences in pathogenicity of the isolates using *N. benthamiana* (Figure 4). These results will contribute to understanding the host range and pathogenic mechanisms of domestic *R. fascians* and will promote further efforts to control the pathogens that cause this disease.

## 3. Materials and Methods

### 3.1. Sample Collection and Bacterial Isolation

Eight symptomatic hybrid lilies (*Lillium* sp.) were collected from greenhouses or nurseries in Gangwon province, South Korea in May 2020 (Appendix A). The collected samples were treated with 1% sodium hypochlorite for 5 min and washed with sterile distilled water. Then, root samples were ground in half-strength phosphate-buffered saline containing 100 μg/mL cycloheximide, 40 μg/mL polymyxin B, and 0.4 μg/mL sodium azide. The extract was filtered through three layers of cheese cloth. The homogeneous extract was allowed to stand for 5 min. Supernatants were serially diluted and plated on modified D2 agar medium [15]. After incubation at 27 °C for 10 days, the orange-colored colonies were selected and restreaked for single-colony isolation. Bacterial isolates underwent a limited number of passages.

### 3.2. Molecular Identification of R. fascians

Orange-colored colonies were inoculated in nutrient broth (MBcell, Seoul, South Korea) and cultured at 27 °C for 2–3 days with shaking. Bacterial DNA was isolated using a Wizard genomic DNA purification kit (Promega Corp., Madison, WI) according to the manufacturer’s instructions for Gram-positive bacteria. For bacterial identification, the 16S rRNA gene was amplified by PCR and analyzed after DNA sequencing. For further identification, the chromosomal virulence locus *vicA* (encoding malate synthase) and *fasD* (encoding isopentenyl transferase) were analyzed by PCR using *R. fascians*-specific primers. The conventional primer sets for *fasD* produced relatively short amplicons of ≈200 bp; therefore, a new primer set was designed using Primer Premier 6.00 to obtain a 573 bp PCR product (Appendix A). To develop a new diagnostic *vicA* primer pair, the full sequence of *vicA* was obtained from 50 *Rhodococcus* species including five pathogenic *R. fascians* strains from publicly available sources, and multiple nucleotide sequence alignment was performed to identify regions that can be used to characterize pathogenic *R. fascians* strains (Appendix A). The specificity of PCR primers was confirmed with total DNA isolated from 10 related *Rhodococcus* species (Figure 2). Primer sets for 16S rRNA, *vicA*, and *fasD* used in PCRs are described in Appendix A. Each 50 μL PCR mixture consisted of genomic DNA (50 ng), 2 μL forward primer (10 μM), and 2 μL reverse primer (10 μM) in 1× nTaq-tenuto PCR premix supplemented with 3’→5′ proofreading activity (Enzynomics Co., Daejeon, South Korea). PCR was performed with the following cycling conditions: an initial denaturation at 95 °C for 3 min, 40 cycles of denaturation at 95 °C for 20 s, annealing at 55 °C for 30 s, and extension at 72 °C for 1 or 1.5 min. A final extension was performed at 72 °C for 10 min.

### 3.3. Phylogenetic Analysis

The PCR products of 16S rRNA, *vicA*, and *fasD* obtained from the bacterial isolates were fully sequenced by overlapping reads in forward and reverse directions using PCR primers and internal primers (785F, GGATTAGATACCCTGGTA and 800R, TACCAGGGTATCTAATCC for 16S rRNA). The resulting sequences were used as queries in BLAST searches (default search parameters) using the NCBI nr database. Based on a similarity of 16S rRNA (≥99%) with the reference sequences, the isolates were considered *R. fascians* (Table 1).

Phylogenetic trees were constructed with the sequences of 16S rRNA, *vicA*, and *fasD* using MEGA 7 software [27] and the neighbor-joining method [28]. The tree is drawn to scale, with branch lengths in the same units as those of the evolutionary distances used to infer the phylogenetic tree. The evolutionary distances were computed using the maximum composite likelihood method. The sequences of reference *R. fascians* strains D188 and A25f were downloaded from the NCBI nr, nt, or wgs database (http://www.ncbi.nlm.nih.gov (accessed on 10 February 2021)) and used to construct the phylogenetic tree. The sequences from the *Corynebacterium glutamicum* strain AS019 (L27123) and nonpathogenic *Rhodococcus* species strain KB6 [5] were used as an outgroup for construction of the phylogenetic tree.

### 3.4. Plant Inoculation Assays

The inoculation of tobacco (*Nicotiana benthamiana*) seedlings was carried out as described previously [24]. Tobacco seeds were germinated on a half-strength MS agar medium containing 1% sucrose. After 3 days of incubation, the seedlings were inoculated with 3 μL *R. fascians* isolates (optical density at 600 nm (OD_600_) of 0.5 in 10 mM MgCl_2_) and vertically maintained at 26 °C under a 16 h light condition. Plant images were taken 7 days after inoculation. For the leafy gall assay of tobacco, apical meristems of 4-week-old plants were pinched using forceps and inoculated with 10 μL *R. fascians* suspension (OD_600_ of 0.5 in 10 mM MgCl_2_). Plant images were taken 3 weeks after inoculation. For a mock treatment, the plant was inoculated with 10 mM MgCl_2_. To isolate *R. fascians* from the infected plants, leafy galls or meristems were excised from plants and ground in half-strength phosphate-buffered saline. Serial dilutions of extracts were plated on nutrient agar plates. After incubation for 3–5 days, orange-colored bacteria were analyzed by PCR using *R. fascians*-specific primers.

### 3.5. Statistical Analysis

Triplicates were performed with five specimens per experiment. Tukey’s honestly significant difference test was used to determine significance using the Student’s t-test (Microsoft Office Excel) and an ANOVA (SPSS v.18; IBM, Armonk, NY, USA). A *p-*value of <0.05 was considered statistically significant.

## Figures and Tables

**Figure 1 pathogens-10-00241-f001:**
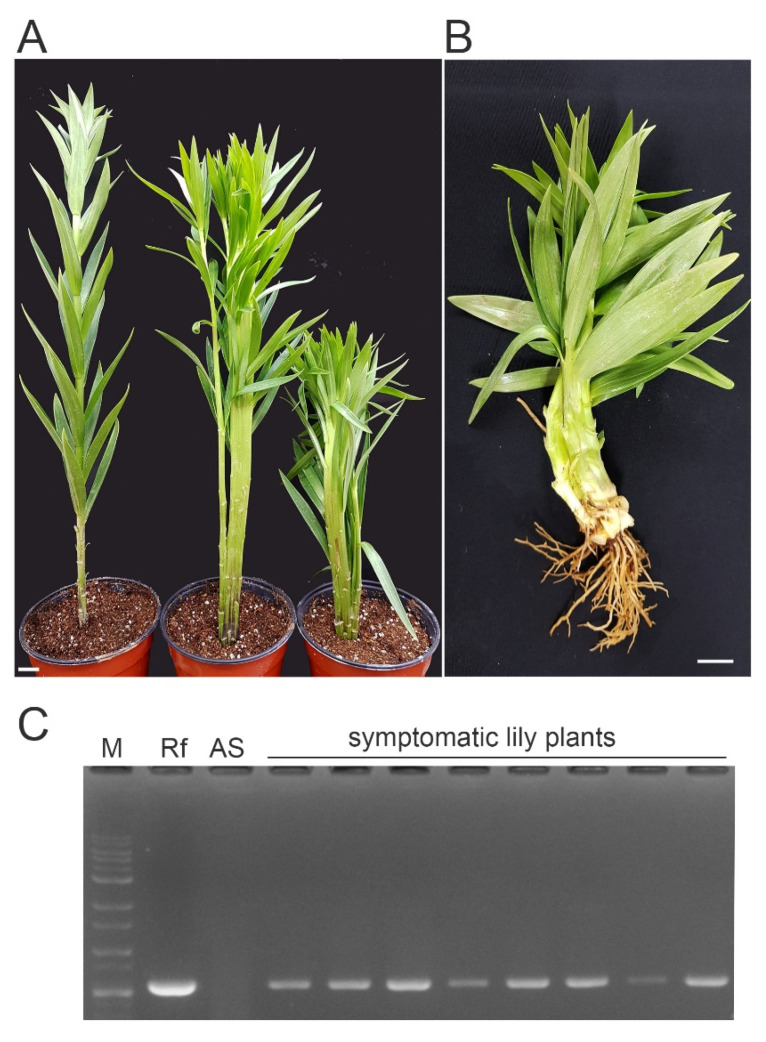
Typical phenotypes of symptomatic hybrid lily plants. (**A**) Multiple inflorescences with flattened stem lilies (middle and right) compared with those of an asymptomatic plant (left). (**B**) Plant exhibiting severe stunted growth with flat twisted stems. Bar, 1 cm. (**C**) PCR amplification of *fasD*. Total DNA from the asymptomatic (AS) and symptomatic lily plants was used as the template. The *fasD* amplicon size is 573 bp. M, molecular weight marker; Rf, positive control for *R. fascians* (Loewe Biochemica GmbH); AS, asymptomatic lily plant.

**Figure 2 pathogens-10-00241-f002:**
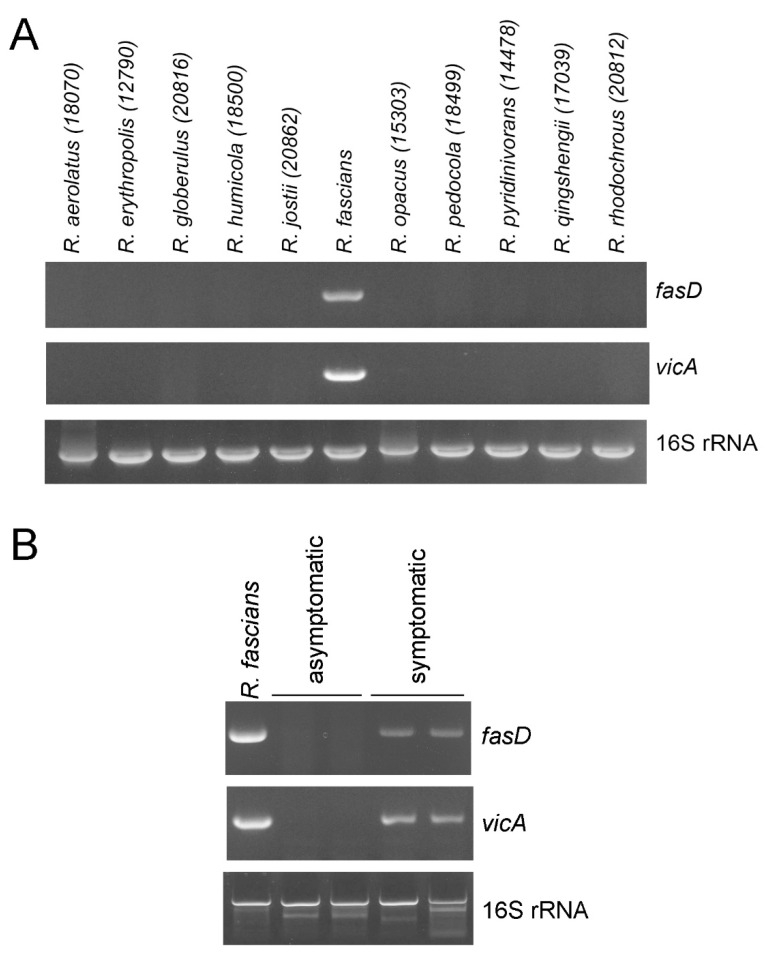
(**A**) Evaluation of PCR primers for *vicA*. Total bacterial DNA was isolated from *Rhodococcus* species obtained from the South Korean Agricultural Culture Collection (KACC) microbial stock center (Rural Development Administration, South Korea). Numbers indicate the KACC accession numbers. (**B**) Bacterial DNA was isolated from asymptomatic and symptomatic lilies. The positive control for *R. fascians* was purchased from Loewe Biochemica GmbH. The *fasD*, *vicA*, and 16S rRNA genes were PCR-amplified using PCR primers (Appendix A). The experiments were repeated at least three times, and representative results are shown.

**Figure 3 pathogens-10-00241-f003:**
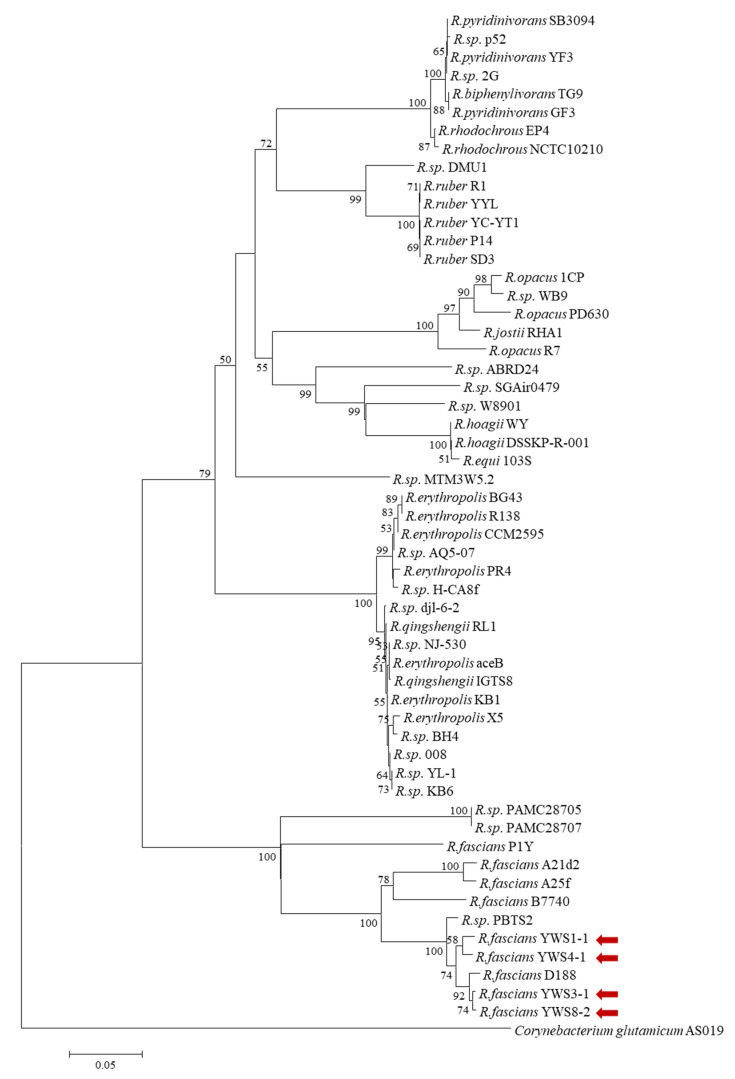
A neighbor-joining tree based on the nucleotide sequence of *vicA*. The *vicA* sequences of 50 *Rhodococcus* species were gathered from the NCBI (National Center for Biotechnology Information) database by a BLASTN search using the 694 bp PCR products obtained from YWS isolates. The node values are bootstrap values generated with 1000 replicates (only values >50% are shown at branch points). The tree was rooted using *aceB* encoding malate synthase of *Corynebacterium glutamicum* AS019 (L27123). Bar, 0.05 substitutions per nucleotide position. Red arrows indicate isolates of *R. fascians* in this study.

**Figure 4 pathogens-10-00241-f004:**
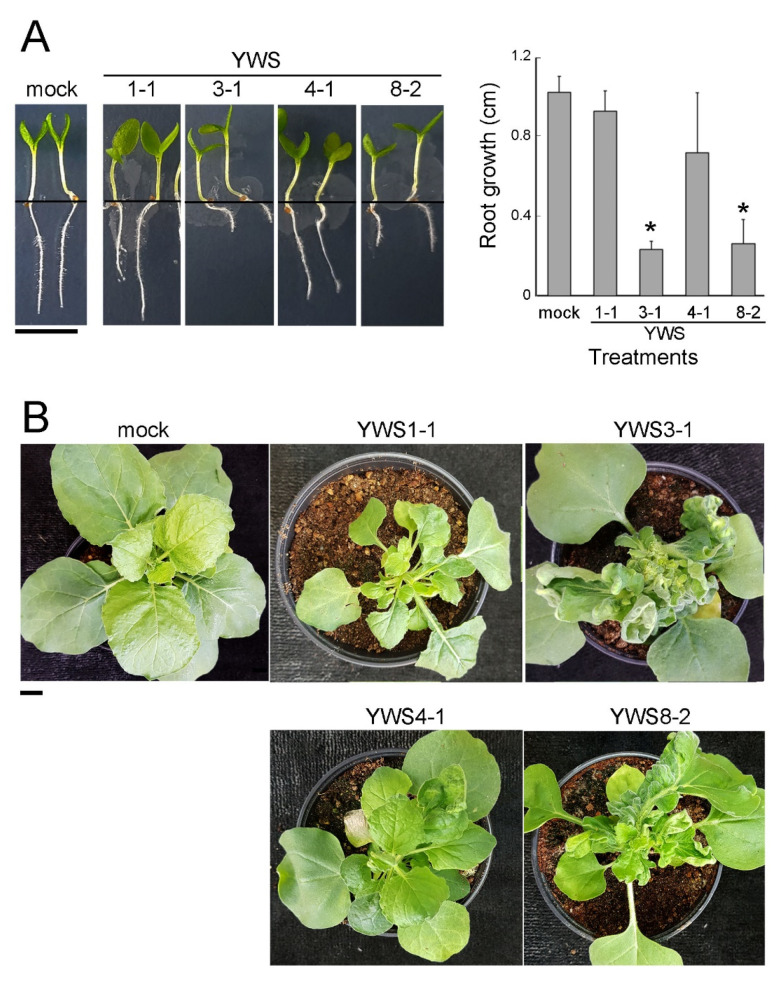
Variable leafy galls syndromes by inoculation of tobacco with YWS isolates. (**A**) Inhibition of *N. benthamiana* root growth by YWS isolates. Seedlings were inoculated with YWS isolates or 10 mM MgCl_2_ buffer (mock), and root lengths were measured 7 days later. Scale bar, 0.5 cm. Error bars indicate standard deviations of the means (*n* = 10); * *p* ≤ 0.01. (**B**) Leafy gall disease of *N. benthamiana* by infection with the isolates. Representative images were taken 3 weeks after inoculation. Scale bar, 1 cm. All experiments were repeated at least three times, with similar results.

**Table 1 pathogens-10-00241-t001:** NCBI GenBank accession numbers for the nucleotide sequences of the 16S rRNA, *fasD*, and *vicA* genes of YWS strains isolated in this study.

Strain	Gene	Accession Number	Length (bp)	Sequence Homology		
				Description	Identity	Accession
YWS 1-1	16S rRNA	MW394216	1518	*R. fascians* D188, complete genome	99.9%	CP015235
	*fasD*	MW394220	573	*R. fascians* D188 plasmid pFiD188	100.0%	CP015236
	*vicA*	MW394212	694	*R. fascians* D188, complete genome	97.8%	CP015235
YWS 3-1	16S rRNA	MW394217	1517	*R.**fascians* D188, complete genome	99.8%	CP015235
	*fasD*	MW394221	573	*R. fascians* D188 plasmid pFiD188	100.0%	CP015236
	*vicA*	MW394213	694	*R. fascians* D188, complete genome	99.1%	CP015235
YWS 4-1	16S rRNA	MW394218	1517	*R. fascians* D188, complete genome	99.4%	CP015235
	*fasD*	MW394222	573	*R. fascians* D188 plasmid pFiD188	100.0%	CP015236
	*vicA*	MW394214	694	*R. fascians* D188, complete genome	96.4%	CP015235
YWS 8-2	16S rRNA	MW394219	1517	*R. fascians* D188, complete genome	99.8%	CP015235
	*fasD*	MW394223	573	*R. fascians* D188 plasmid pFiD188	100.0%	CP015236
	*vicA*	MW394215	694	*R. fascians* D188, complete genome	99.0%	CP015235

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
