# Peer review of "Detection of *Rhodococcus fascians*, the Causative Agent of Lily Fasciation in South Korea"

_pathogens, 2021, doi:10.3390/pathogens10020241_

Round 1
Reviewer 1 Report
This Manuscript by Park et al sampled symptomatic lily plants in Gangwon province, South Korea and performed a molecular identification (PCR and amplicon sequencing) and symptomatology study to confirm that the symptoms were caused by isolates of Rhodococcus fascians. R. fascians is an economically important pathogen causing economic losses in lily cultivation, and a better identification protocol can help in minimizing the losses.
Authors mention that they developed a new PCR protocol to differentiate pathogenic isolates based on the chromosomally encoded vicA gene. I am not fully convinced with this approach as the plasmid borne virulence factors seem to be essential for pathogenicity. It appears that the vicA gene is carried even by the non-pathogenic isolates (lines 137-143), and only the isolates carrying the plasmid (i.e. fasD positive) are virulent. As such, I am not quite sure of the usefulness of this new protocol. Given the fact that vicA based approaches are already available for detection (lines 89-90, reference #18), and fasD primers seem to be already available (see my comment below), the novelty of this work is in confirming that R. fascians is found in South Korea (is this the first time this being reported?). I think the writing needs to be improved to convey this message clearly.
Line 27. I think Actinobacteria is understood more as a phylum than a class? How about re-writing this sentence as ‘The genus Rhodococcus contains Gram-positive aerobic bacterial species belonging to the phylum Actinobacteria’?
I could not tell if fasD based PCR was designed in this study. Table S2 says the primers were designed in this study, but line 43 and lines 71-73 suggest fasD PCR has been used previously. Also, vicA based primer design is described in detail in the results 2.2, fasD is not described at all. If fasD primers were designed in this study, similar discussion is required.
Lines 137-143. So the isolates giving vicA positive amplicons are still non-pathogenic? This suggest that having the plasmid is essential for virulence, and an amplification scheme targeting the vicA gene only may not be able to tell if there is a pathogenic or non-pathogenic isolate?
Figure 3, I think neighbor joining is not a best approach for phylogeny, although it might serve the purpose here. The figure legend needs little more about what the line indicates, how many bootstrap replications, outgroup used etc.
Also, mention in the methods, were same PCR primers used for sequencing, were both forward and reverse sequenced in full, how long was the sequence used for phylogeny? How were the sequences aligned? These are minor details but need to be included in the methods for studies like this which are based on identifying pathogens by sequencing.
Lines 208. Does 1× nTaq-tenuto PCR premix high fidelity?
Reviewer 2 Report
The work entitled “Identification of Rhodococcus fascians, the causative agent of lily fasciation in South Korea”, authors are Joon Moh Park, Jachoon Koo, Sewon Kang, Sung Hee Jo, and Jeong Mee Park, is devoted to identification and characterization of plant pathogen Rhodococcus fascians. This phytopathogen is not properly studied, and the work done adds into knowledge about R. fascians intraspecific variance and plant disease symptoms. The work is logical and well-structured. The pathogen isolates are obtained in pure cultures, and their pathogenicity (virulence) is estimated. The experiments are properly done with all necessary controls. Novel and important data is development of primers for the detection of chromosomal vicA gene specifically in phytopathogenic Rhodococcus.
Major revisions
- The aim of the work is postulated by authors as R. fascians identification in symptomatic lilies but limited number of tests is used for this purpose including only colony morphology description and 16S rRNA, vicA, and fasD gene sequencing. Identification of bacteria on the base of the 16S rRNA gene is not precise at the species level (Chun et al. Proposed minimal standards for the use of genome data for the taxonomy of prokaryotes // IJSEM. 2018. 68:461–466). The genes vicA and fasD are functional and suitable to build phylogeny trees and understand evolution of pathogenic traits but functional genes are not recommended for taxonomy. In the context of polyphasic taxonomy for proper identification of isolates, whole genome sequencing or at least MLSA (sequencing of 7 conserved genes) plus analysis of phenotype should be performed (Chun et al. Proposed minimal standards for the use of genome data for the taxonomy of prokaryotes // IJSEM. 2018. 68:461–466).
- Authors have analyzed only 8 plants with detection of R. fascians in four. It is little for adequate statistics.
Minor revisions
- 5, lines 118–120 – Were any bacteria (R. fascians or other) isolated from other four symptomatic lily plants? Or their symptoms were related to another reason not bacterial/R. fascians infection?
- 5, line 126 – R. fascians, not R. fasciens.
- 5, line 137 – it is recommended to write “rhodococci” (start from lower-case letter, no italics) not “Rhodococci” since it is trivial not a taxonomic Latin name.
- 8, lines 208–209 – Is DNA-polymerase of the nTaq-tenuto PCR premix high-fidelity? For sequencing, nucleotides in PCR-products should be identical (minimum of errors at DNA copying with a DNA-polymerase) to nucleotides in native genes.
Summary. Overall recommendation: Reconsider after major revision.

Reviewer 3 Report
Authors performed examination of isolates Rhodococcus fascians, the causative agent of fasciation in lily plants grown in South Korea. Futhermore phylogeny analysis with nucleotide sequences leds to the classification of the isolates into four different strains of R. fascians.
The manuscript provides knowledge about genetic diversity of pathogenic R. fascians and may be useful for researchers in this field.
However, in my opinion, combining the results and the disscusion into one section makes manuscript difficult to read. Please consider to divide Results and Discussion into two sections. Futhermore conclusions seems to be too general. Could you clarify more findings of your study?
Futhermore I have 2 minor issues which should be corrected/considered:
1.reference no 25: please use abbreviated name of journal
2.line 245: please clarify which statistical software was used
Round 2
Reviewer 1 Report
Thank you again for this opportunity to review this paper. I think the authors were really good in responding to the comments.
Reviewer 2 Report
The manuscript could be accepted in its current form. However, it is recommended to replace "identification" with "detection" in the title. The title "Detection of Rhodococcus fascians, the causitive agent of lily fasciation in South Korea" reflects the article content more than the title "Identification of Rhodococcus fascians, the causitive agent of lily fasciation in South Korea". Identification means using several relevant taxonomic methods, while for detection, full information is provided in the article.